# Associations between Fundamental Movement Skills and Moderate-to-Vigorous Intensity Physical Activity among Chinese Children and Adolescents with Intellectual Disability

**DOI:** 10.3390/ijerph192013057

**Published:** 2022-10-11

**Authors:** Taijin Wang, Yingtao Qian, Tianwei Zhong, Jing Qi

**Affiliations:** College of Physical Education and Health Sciences, Zhejiang Normal University, Jinhua 321004, China

**Keywords:** motor skills, locomotor skills, object control skills, physical activity, intellectual disability

## Abstract

Higher physical activity (PA) levels will obtain more health-related benefits for children and adolescents with intellectual disabilities (ID). The mastery of fundamental movement skills (FMS) potentially correlates with PA. This study aimed to examine the associations of FMS with moderate-to-vigorous intensity physical activity (MVPA) levels in children and adolescents with moderate to severe ID. Moreover, this research analyzes whether there are gender and age differences in the association between these two variables. A total of 93 children and adolescents with ID, aged 8–17 years (mean age = 13.27; SD = 3.35), were recruited from a special school located in western China. The time spent in MVPA was measured using waist-worn accelerometers. FMS proficiency was assessed using the Test of Gross Motor Development 2 (TGMD-2). Children and adolescents with ID tend to have delayed maturity of FMS patterns (locomotor skills *t* (92) = −16.91, *p* < 0.001, *d* = 2.48; object control skills *t* (92) = −25.39, *p* < 0.001, *d* = 3.72; total FMS *t* (92) = −21.83, *p* < 0.001, *d* = 3.20) and lower proficiency in objective control skills (*t* (92) = 3.989, *p* < 0.001, *d* = 0.29). A significant positive correlation was found between MVPA and FMS, and this association was moderated by gender and age. For boys, object control skills were a significant predictor of MVPA time (*B* = 0.842, *p* < 0.01), whereas locomotor skills were a significant predictor of MVPA time (*B* = 0.472, *p* < 0.05) for girls. For children with ID, object control skills were a significant predictor of MVPA time (*B* = 0.736, *p* < 0.05). Proficiency in FMS has a positive effect on increasing the level of MVPA in children and adolescents with ID. Gender and age factors should be considered when implementing FMS intervention programs.

## 1. Introduction

Higher physical activity (PA) levels are associated with physiological, psychological, and psychosocial health among children and adolescents [1,2,3]. In particular, an increase in moderate-to-vigorous-intensity physical activity (MVPA) has beneficial effects on cardiometabolic risk factors [4] and improves cognition among children and adolescents [5,6,7]. The World Health Organization (WHO) thus recommends that children and adolescents, with and without disabilities, should engage in at least 60 min of MVPA daily to obtain related health benefits [8]. However, owing to the presence of disability and associated conditions, youth with disabilities are less active than their counterparts without disabilities, and only low proportions of them meet the WHO recommendation for PA [9,10]. Therefore, youth with disabilities are often at a greater risk for health problems [11,12]. Given the significant influence of PA on an individual’s health, investigating potential factors that influence MVPA levels among children and adolescents is important, particularly regarding those who are at the greatest risk of being physically inactive [13].

An essential contributor to the amount of PA children and adolescents take part in is fundamental movement skill (FMS) competence [14]. FMS is the ability to coordinate the use of basic motor skills, which are considered the building blocks that lead to specialized movement sequences required for adequate participation in many organized and non-organized physical activities for children, adolescents, and adults [15]. FMS is usually categorized as either locomotor (e.g., running, jumping, and hopping) or object control (e.g., throwing, catching, and kicking) [16]. Children and adolescents with better developed FMS choose to participate in a wider range of PA, allowing them to gain enriching sporting experiences that can improve their PA levels [17]. Accumulating evidence shows that FMS is positively associated with PA levels in youth without disabilities [18,19]. In addition, the moderating effect of gender and age on the association between FMS and PA is proven [17,20,21,22]. For example, Bryant et al. (2014) [20] showed that girls’ previous FMS proficiency was a better predictor of future PA levels compared to boys. Jaakkola et al. (2019) [21] found that significant associations between FMS and PA appear only for girls. The results of a review by Logan et al. (2015) [22] show a stronger correlation between FMS and PA during childhood than during adolescence. However, evidence pertaining to youth with disabilities is limited [23,24,25], especially for children and adolescents with intellectual disabilities (ID).

ID is defined as limitations both in intellectual functioning and adaptive behavior and originate before the age of 18 years; those with ID constitute about 1–4% of the global population [26]. Children and adolescents with ID tend to have deficits in cognition, language, psychosocial, and motor proficiencies [27,28]. Children and adolescents with ID report lower levels of PA when compared with their peers without disabilities [10,29]. Eguia et al. (2015) [24] examined the relationship between FMS and PA in Philippine children with ID using pedometers, and they found that object control skills in FMS are a significant predictor of daily step counts. However, pedometers are susceptible to abnormal gait patterns, which are common among children and adolescents with ID [30]. Compared with pedometers, accelerometers measure the acceleration of the body during movement, which can provide the most comprehensive and feasible method for measuring PA parameters among children and adolescents with ID [31]. Despite Eguia’s study, further evidence of FMS related to PA among adolescents with ID are still needed, as their study mainly focused on children with ID and may not be generalized well for youth with ID.

Therefore, using accelerometers to measure PA levels and the Test of Gross Motor Development 2 (TGMD-2) to assess FMS proficiency, the objectives of this study are (1) to assess whether there are associations between FMS and MVPA levels of children and adolescents with ID, and (2) to analyze whether there are gender and age differences in such an association. We hypothesize that (1) there is significant association between FMS and MVPA levels of children and adolescents with ID, and (2) the association between FMS and MVPA is moderated by gender and age.

## 2. Materials and Methods

### 2.1. Participants

We used a non-probability sample method and recruited 142 Chinese children and adolescents with ID from a special school for students with developmental disabilities in Yinchuan, a city in western China. This school includes primary school (grades 1–6, aged from 7 to 12 years) and middle school (grades 7–9, aged more than 13 years old). There are two classes per grade level in primary school and one class per grade level in middle school. Each class comprises 10–15 students. The school classifies students based on ID level information on the disability certificate, which is determined by doctors designated by the Federation of Individuals with Disabilities in China in strict accordance with disability identification standards. The disability identification standards use IQ and adaptive behavior to identify ID and classify the levels of ID, which are consistent with the WHO and the American Association on Intellectual and Developmental Disability (AAIDD) classification criteria for ID. Participants must meet the following criteria to be eligible for the study: (a) aged 5–17 years, (b) previously diagnosed with significant limitations in intellectual functioning and adaptive behavior, (c) have no other physical disability or health condition that may limit musculoskeletal movements (e.g., cerebral palsy), (d) possess sufficient verbal communication skills to follow instructions and complete all the tests independently [32], without behavioral problems that may affect the performance of instructions (e.g., oppositional behavior) [24], (e) have completed the full TGMD-2 test program and had at least three days (two school days and one weekend day) of valid accelerometer data, and (f) be present during the investigation. According to the criteria, 49 students were excluded due to the following reasons: (a) 7 students had other physical disabilities or health conditions that limit musculoskeletal movements, (b) 12 students were unable to follow instructions to complete all tests of TGMD-2 independently, (c) 3 students had behavioral problems that interfered with the execution of instructions, (d) 23 students’ accelerometer data did not cover at least three valid days, and (e) 4 students withdrew during the test. The final analytic sample consisted of 93 participants (8–17 years old, M age = 13.27, SD = 3.35); 63 (67.74%) were boys and 30 (32.26%) were girls.

### 2.2. Measurements

Background information. The student affairs office of the school provided the background information, including gender, age, grade, and ID level of each participant. The age groups were defined. The participants were classified into two groups in accordance with the age division of the primary and middle school of this special school and the WHO classification of age groups [33]. The age classifications were Age Group 1, which included students aged 7–12 years (*n* = 49) in the primary school department, and Age Group 2, which included students aged 13–17 years (*n* = 44) in the middle school department.

Anthropometric Measures. During the school visits, we measured the height and weight of each student. Height (cm) was measured to the nearest 0.1 cm in bare feet, whereas weight (kg) was examined to the nearest 0.1 kg using GMCS-IV (Jianmin, Beijing, China).

PA Measurement. The PA of children and adolescents with ID was measured using the ActiGraph wGT3-BT accelerometer. This device translates movement in the direction of three internal axes into counts and has been validated among children and adolescents with ID [31]. Before the test, acceleration is initialized, and the time for starting and stopping data recording is set. Each participant was instructed to wear the monitors on his or her waist by using one of the compatible belts for seven consecutive days while awake and remove it while bathing, swimming, or sleeping. The sampling interval (epoch) in the study was set to 15 s, which has been used in studies assessing PA levels of children and adolescents with ID [29,34,35], and presented acceptable reliability and validity [36]. Thus, accelerometer data were output in counts/15 s. After the test, the original ActiGraph data were processed with the ActiLife Lifestyle Monitoring System (software version 6.13.4) and filtered for validity. To include a larger number of participants in data analyses, the requirement of valid physical activity data was set as a minimum of three days (two school days and one weekend day), with at least 10 h wearing time per day. Three days of accelerometer-based data produced a reliability coefficient of 0.7, which is considered acceptable and justified [37]. Periods ≥ 10 min with zero counts are considered as non-use time and excluded from the analysis [38]. MVPA was defined when the cut point was ≥2296 counts per minute. This threshold was first established by Evenson et al. (2008) [39] and later independently validated by Trost et al. (2011) [40]. To obtain the average time per day spent on each level of PA, the total time (min) for that intensity level was divided by the number of days for each participant. If they met the WHO standard of at least 60 min of MVPA daily, participants were judged to have met the recommended amount [8].

FMS Assessment. The TGMD-2 was used to measure FMS of the participants [41], which included both locomotor skills and object control skills, and has been validated in children and adolescents with ID [42,43]. Although originally designed for assessing children aged 3~10 years, this instrument still showed acceptable reliability and validity involving youth aged from 11–18 years [42,44]. The TGMD-2 examines the quality of movement patterns based on a number of qualitative criteria (3~5 for a specific skill). The presence or absence of a criterion is scored 1 or 0, yielding a maximum score of 3~5 per trial. In this study, 24 points were assigned to six locomotor components, including the ability to: run (4 points), gallop (4 points), hop (5 points), leap (3 points), horizontally jump (4 points), and slide (4 points), whereas the other 24 points were assigned to six object control skills: strike a stationary ball (5 points), stationary dribble (4 points), catch (3 points), kick (4 points), overhand throw (4 points), and underhand roll (4 points). Additionally, all tests were conducted twice for each the participant. Given the presence of disability in our samples, we used raw scores of FMS for data analysis instead of the normative values which are usually used for youth with typical development [24,45,46]. Thus, the subtest scores were calculated by adding together the scores for each of the six locomotor skills (maximum possible score = 48) and the six object control skills (maximum possible score = 48) included in the TGMD-2 protocol, respectively. To quantify mastery in each sub-skill component of FMS, the following approach used in recent studies to analyze TGMD-2 scores was adopted [47,48,49,50]. ‘Mastery’ was defined as the correct performance of all criteria over two trials (e.g., score of 8 for the run). ‘Near-mastery’ was defined as the correct performance of all but one performance criteria over two trials, which for the run was assigned a score of 7 (A score of 6 would also be classified as near mastery if only one criterion was absent over two trials). ‘Poor’ was defined as the incorrect performance/absence of more than one performance criterion over two trials.

For subjects with severe ID, we conducted a familiarization protocol to ensure the participant gave his or her best effort to perform the movements. This protocol involved the following steps: (1) Each movement was demonstrated to the participants twice; (2) The participants practiced the movement twice with prompting; (3) The participant practice without prompting; and (4) The participant performed the movement twice without prompting, with the best score used for data analysis.

### 2.3. Procedures

The study protocol was approved by the Ethics Committee of Zhejiang Normal University (ZSRT2021075). The principal and PE teachers of the school were contacted in advance to inform them of the project and ask for permission from the school to conduct this study. Prior to data collection, information on the purpose of the study, what would be tested, and the safety of the devices worn was explained to the participants’ parents or guardians. An informed consent form was distributed. After completing the anthropometric measures, we uniformly distributed and attached the accelerometers to each participant, explained to teachers and guardians how the participants should wear the accelerometers and explained the precautions, and asked them to return the accelerometers when data collection was completed. Participants did not need to change their normal daily routines during the monitoring period. We reminded teachers in charge of participant classes to pay more attention to the accelerometers worn by the participants. During the test, we visited the school each day to check the participants’ accelerometer wear. PA measurements were followed by assessments of FMS. In consulting with the school, it was determined that the test should take place during extracurricular activities so as not to disrupt the normal school day. We carried out the tests in collaboration with teachers. Before performing each test, we explained and demonstrated the test item to the participants twice, and the participants tried to imitate it. The intelligence and attention span of children and adolescents with ID are limited [43]. To enable the participants to perform the test movements accurately, the researcher again explained and demonstrated the item individually to each participant before the test. Afterward, the participant completed the item twice independently. We used language such as “very good” and “great” to motivate the participants during the test. To obtain accurate scoring results, two researchers scored one participant simultaneously, and the average of the two raters was the final score. The test procedures were carried out by one researcher and three assistant researchers. All researchers underwent rigorous training, and an interobserver agreement of 85% was used as the criterion to start the assessment. The research process is shown in Figure 1. The data were collected in April 2021, and all measurements or assessments were completed within 10 days.

### 2.4. Data Analyses

Descriptive (mean and standard deviation) and normality statistics (skewness and kurtosis) were calculated for all measures to check the distribution of the data. Since data were normally distributed, the following statistical methods were adopted. The multiple analysis of variance (MANOVA) test was used to test the gender differences among variables and the differences between age groups. Eta-squared (*η2*) was used to provide estimates of effect size (ES). The following values, as outlined by Cohen [51], were used to define ES magnitude: 0.01–0.06 (small effect), 0.06–0.14 (medium effect), and >0.14 (large effect). Comparisons of demographic characteristics were determined by independent sample *t* test. The difference between locomotor skills and object control skills was analyzed by paired sample *t* test. A one-sample *t* test was used to test the difference between the observed scores of locomotor skills, object control skills, and total FMS and their full scores. The ES of the *t* test was calculated according to Cohen’s *d* statistic, exercising the following thresholds: <0.2 (trivial effect), 0.2–0.59 (small effect), 0.6–1.19 (moderate effect), 1.20–1.69 (large effect), and >1.70 (very large effect). Pearson’s correlation coefficient was used to test the correlation between FMS and MVPA time; hierarchical regression analysis was performed to test the association between FMS and MVPA in children and adolescents with ID, along with the possible gender or age interaction. Two hierarchical regression models were performed to specify the variable that contributed to the MVPA by gender or age, if gender or age interaction were observed. Each regression tested three models. Participants’ age or gender were controlled when Model 1 examining gender or age contribution to MVPA was performed. Locomotor skills and objective control skills were added in Model 2. Finally, the interactions between variables (e.g., interaction between age and locomotor skills) were also added. Statistical significance was set at *p < 0.05* for all tests. SPSS 26.0 (IBM, Armonk, NY, USA) was used for data analysis.

## 3. Results

### 3.1. Demographic Information of the Participants

The demographic information, including gender, age, height, and weight of the participants, is shown in Table 1. No significant differences were observed between boys and girls for age, *t*(92) = 0.268, *p* = 0.789, *d* = 0.06; height, *t*(92) = 1.345, *p* = 0.182, *d* = 0.31; or weight, *t*(92) = 1.856, *p* = 0.067, *d* = 0.39.

### 3.2. FMS Proficiency and MVPA Time

Table 2 shows the mastery levels of FMS in subjects with ID. The proportion of participants who displayed mastered FMS was low, ranging from 0% (overhand throw) to 21.51% (gallop). The proportion of near-mastery FMS ranged from 4.30% to 48.39%. Over half of the participants showed poor mastery in eleven of the twelve skills, including overhand throw (95.70%), underhand roll (91.40%), kick (86.02%), horizontal jump (80.65%), hop (80.65%), strike a stationary ball (77.42%), leap (76.34%), run (75.27%), catch (66.67%), stationary dribble (66.67%), and gallop (59.14%).

Table 3 presents the descriptive statistics of all outcome variables. One-sample *t* test showed that the mean scores of the participants obtained were significantly lower than the full score for locomotor skills, 27.37 (11.77), *t*(92) = −16.91, *p* < 0.001, *d* = 2.48, object control skills, 24.29 (9.01), *t*(92) = −25.39, *p* < 0.001, *d* = 3.72, and total FMS, 51.66 (19.59), *t*(92) = −21.83, *p* < 0.001, *d* = 3.20. The mean scores obtained by boys for locomotor skills, object control skills, and total FMS were 28.17 (11.49), 25.37 (7.80), and 53.54 (18.25), respectively. The mean scores obtained by girls for locomotor skills were 25.67 (12.36), object control skills were 22.03 (10.93), and total FMS were 47.70 (21.95). The mean scores obtained by children with ID for locomotor skills were 26.00 (11.97), object control skills were 23.22 (8.74), and total FMS were 49.22 (19.48). For adolescents with ID, the mean scores obtained for locomotor skills, object control skills, and total FMS were 28.89 (11.49), 25.48 (9.28), and 54.36 (19.60), respectively. Paired sample *t* test results showed that for the total sample of participants, (*t*(92) = 3.989, *p* < 0.001, *d* = 0.29); for boys, (*t*(62) = 3.076, *p* < 0.01, *d* = 0.29); and for girls, (*t*(29) = 2.517, *p* < 0.05, *d* = 0.31) in this study, significantly higher scores were obtained for locomotor skills than for object control skills. In addition, the locomotor skills scores remained significantly higher than the object control skills scores among children (*t*(48) = 2.519, *p* < 0.05, *d* = 0.38) and adolescent (*t*(43) = 3.148, *p* < 0.01, *d* = 0.31).

Overall, the participants with ID in the study engaged in an average of 39.73 (12.50) min of MVPA per day. Only 9.68% of children and adolescents with ID met the WHO recommendations of 60 min of daily MVPA.

Table 4 shows the results of MANOVA for FMS and MVPA by gender and age. No significant gender or age differences were observed in the locomotor skills (gender *F*(1,91) = 0.996, *p* = 0.321, *η2* = 0.011; age *F*(1,91) = 1.403, *p* = 0.239, *η2* = 0.016), object control skills (gender *F*(1,91) = 3.159, *p* = 0.079, *η2* = 0.034; age *F*(1,91)= 0.551, *p* = 0.460, *η2* = 0.006), and total FMS scores (gender *F*(1,91) = 1.990, *p* = 0.162, *η2* = 0.022; age *F*(1,91) = 1.111, *p* = 0.295, *η2* = 0.012) of children and adolescents with ID. No interaction was found between gender and age for locomotor skills (*F*(1,91) = 0.019, *p* = 0.891, *η2* = 0.000), object control skills (*F*(1,91) = 1.714, *p* = 0.190, *η2* = 0.019), or total FMS scores (*F*(1,91) = 0.265, *p* = 0.608, *η2* = 0.003). No significant gender (*F*(1,91) = 2.660, *p* = 0.106, *η2* = 0.029) or age (*F*(1,91) = 0.459, *p* = 0.500, *η2* = 0.005) differences in MVPA time were observed, and no interaction was found between gender and age for MVPA time (*F*(1,91) = 0.189, *p* = 0.665, *η2* = 0.002).

### 3.3. Association between FMS of Children and Adolescents with ID with their MVPA Time

Table 5 presents the bivariate Pearson product-moment correlations in the MVPA time of children and adolescents with ID regarding locomotor skills, object control skills, and the general FMS by gender and age. All study variables were positively correlated.

Table 6 presents the results of the regression analysis for MVPA time, regressed by gender, age, and FMS variables, among participants. In Model 1, age was a significant negative predictor of MVPA time (*B* = −0.401, *p* < 0.05), whereas gender was not a significant predictor (*B* = 4.628, *p* = 0.095). Locomotor skills and objective control skills were entered in Model 2 (after controlling for gender and age) and significantly accounted for 37.1% of the variance in MVPA time of children and adolescents with ID. Object control skills were a significant predictor of MVPA time (*B* = 0.534, *p* < 0.05) in the total sample. The adjusted model was rerun separately by gender. For boys, object control skills were a significant predictor of MVPA time (*B* = 0.842, *p* < 0.01). For girls, locomotor skills were a significant predictor of MVPA time (*B* = 0.472, *p* < 0.05). Similarly, the adjusted model was rerun separately by age. For children with ID, object control skills were a significant predictor of MVPA time (*B* = 0.736, *p* < 0.05). For adolescents with ID, none of the skill variables (locomotor skills *B* = 0.377, *p* = 0.086; object control skills *B* = 0.322, *p* = 0.254) were significant predictors of MVPA time (Table 7).

## 4. Discussion

This study used device-based data to examine the association between FMS and MVPA time of children and adolescents with ID. To the best of our knowledge, this study is the first to investigate the contribution of FMS to the daily MVPA of children and adolescents with ID.

Our study revealed that most of the participants showed low levels of FMS proficiency, the proportion of mastery proficiency in sub-skills ranging from 0% to 21.51%. This finding is consistent with the observations regarding examining FMS proficiency of children and adolescents with ID (e.g., Capio et al. (2010) [52]; Schott and Holfelder (2015) [53]) and comparing FMS proficiency between children and adolescents with ID and their peers without disabilities (e.g., Rintala et al. (2013) [54]; Alesi et al. (2018) [55]). Children and adolescents with ID can perform simple movements. However, when more parts of the body are simultaneously involved in the skills (e.g., striking a stationary ball and horizontal jumping), it causes problems for children and adolescents with ID. Poor cognitive proficiency of children and adolescents with ID is the most likely reason for their low FMS performance [55,56]. In addition, children and adolescents with ID in this study showed lower proficiency for objective control skills, such as overhand throwing and underhand rolling, than for locomotor skills, such as sliding and galloping. This is consistent with results from previous studies [47,55]. These differences may also be attributed to the lower levels of cognitive functioning in children and adolescents with ID [57]. Performance on tasks such as throwing, catching, and striking requires a higher cognitive load and a more sophisticated cognitive process related to goal-directed behaviors and executive functions [57,58]. Nevertheless, locomotor tasks, such as walking, running, and hopping, involve stereotypic movements that encompass automatized cognitive functioning [23]. Findings from a recent literature review regarding synthesizing studies of FMS in youth with ID indicate that children and adolescents with ID with higher levels of executive or cognitive function (processing speed and verbal comprehension) tend to have significantly higher performance for locomotor and objective control skills, or FMS in general [59]. The impacts of ID on FMS performance should be examined in future studies.

This study showed that the mean daily MVPA of the participants was 39.73 min, which is lower than the WHO’s recommended MVPA level. This is consistent with the findings of previous studies; that is, the global PA levels of children and adolescents with ID do not reach international PA recommendations (e.g., those of the UK, Iceland, the US, and the Netherlands) [34,60,61,62]. Three reasons may explain the low MVPA levels of Chinese children and adolescents with ID observed in the current study. First, due to the limitations in physical, cognitive, and social skills, the ‘mainstream’ organized PA and/or sporting opportunities are usually not appropriate for children and adolescents with ID. A general lack of PA and/or sporting opportunities designed specifically for this population makes it rare for them to engage in PA in their leisure time; thus, they spend most of their time in a sedentary state [63]. When they engage in PA, the duration of PA also decreases as the intensity increases [64], and low-intensity walking may be the dominant form of exercise [65]. Second, children and adolescents with ID in this study may have lower PA levels outside of school because they may spend too much time watching TV or playing computer games during after-school hours. The special school attended by the participants is a non-boarding school. Following the school’s schedule, all students leave school at 4:30 pm every day. Participants may have sedentary lifestyles when they return home. Additional reasons may be related to parents’ overprotection. Influenced by Confucianism, Chinese parents raise their children, especially those with disabilities, with a preference for practical control over their them [66]. Parents’ high levels of overprotection and concerns relating to their child’s competence for participating in various PAs may prevent their child from engaging in MVPA.

FMS significantly explained 32.8% of the variance in the MVPA of children and adolescents with ID in this study. The explained variance in the percentage of MVPA duration of children is similar to those reported in other studies focusing on children and adolescents without disabilities, which ranged from 5% to 39% [20,21,67,68], and children and adolescents with disabilities (e.g., cerebral palsy, 76.4% [23]). Developmental delay in FMS may make children and adolescents with ID less willing or less likely to be involved in PA [59]. FMS is believed to improve PA levels in children and adolescents by improving their extended motor repertoire [69] and providing opportunities to for them to participate in a variety of sports and games [13]. However, a large proportion of the variance in MVPA of children and adolescents with ID remained unexplained, suggesting that the influence of FMS (locomotor skills and object control skills) on MVPA is limited. In addition, the MVPA of children and adolescents with ID may be shaped by numerous other factors, such as parental support and engagement [70,71], PA patterns [64], and opportunities to participate in sports activities [72]. Therefore, further studies must include more factors to explain the MVPA of Chinese children and adolescents with ID.

The findings indicated that boys’ object control skills were significantly related to their MVPA time, whereas girls’ locomotor skills were significantly related to their MVPA time. These results supported previous studies [20] demonstrating that boys’ object control skills in the first year predict more of the variation in PA in the second year, and that girls’ locomotor skills in the first year predict more variation in PA in the second year. Locomotor skills are the foundation of physical fitness, whereas object control skills are the foundation of skill-based sports. Sports based on object control skills are characterized by their larger numbers, relative difficulty to master, and requirement of long-term practice [47,57,73]. Compared with girls, boys are more likely to participate in ball sports [74]. Therefore, it is essential for boys to acquire object control skills, such as catching, to participate in these organized sports. Locomotor skill competence may be more important than object control skill competence for girls, as they may be engaging in types of PA (e.g., dance and gymnastics) that do not require object control skills. Therefore, gender differences should be considered when implementing FMS interventions. Object control skill training is a potential contributor toward helping boys to become more physically active. The development of locomotor skills may remain an important area of focus within girls’ school physical education curriculum and extracurricular activities [21].

The findings indicated that children’s object control skills were significantly related to their MVPA time, but none of the skill variables were significantly related to the MVPA time of adolescents with ID. Middle childhood (5–12 years) is considered an important period for creating positive or negative PA trajectories [75,76]. During this period, children participate in an increasing number of team sport activities which require high levels of object control skills [68]. Children with more proficient object control skills may dominate these sports, thus increasing their PA levels [13]. Adolescents with ID rely on participation in more structured PA than children with ID [62], which makes them more susceptible to external factors such as parental support when engaging in PA [77]. Therefore, locomotor skills and object control skills may not directly predict the PA level of adolescents with ID. We suggest implementing FMS intervention programs focusing on object control skills. Such programs can potentially contribute toward increasing the health benefits that are related to heightened PA engagement [24]. Further research is needed to explore the association between FMS and PA levels, particularly in adolescents with ID.

### Strengths and Limitations

This study is characterized by several important strengths. One is that the degree of ID of the participants recruited in this study was concentrated in moderate-to-severe ID. Further strengths of our study are the use of device-based instruments to analyze MVPA. However, this study had certain limitations that must be noted. First, despite our relatively large sample size compared to prior studies [24,25], our findings were based on students from one special school. Thus, the results may not be generalized to other children and adolescents with ID from other regions. A large sample of multiple schools should be targeted in future studies. Second, as a cross-sectional survey design was used, the reported causal relations should be cautiously interpreted. Longitudinal research is needed to establish any sort of causal relationship between the FMS and PA of children and adolescents with ID. In addition, more research is needed to reaffirm whether improved FMS can result in increased PA. Finally, although our device-based evaluation of PA was advantageous in several ways, accelerometry does not reveal PA type or context, both of which may also be important in improving PA levels in children and adolescents with ID. In addition, the cut points of accelerometers are prone to bias; thus, the validity of these cut points specific for children and adolescents with ID require further examination.

## 5. Conclusions

In conclusion, most of the participants showed low levels of FMS proficiency, the proportion of mastery proficiency in sub-skills ranging from 0% to 21.51%. FMS proficiency of the participants is positively associated with the MVPA time, and object control skills were a more significant predictor of MVPA than locomotor skills. Therefore, the FMS intervention to promote the MVPA of Chinese children and adolescents with ID is important, especially in developing and improving object control skills. Additionally, the associations between FMS and MVPA were moderated by gender and age. Specifically, boys’ object control skills were significantly associated with their MVPA time, whereas girls’ locomotor skills were significantly associated with their MVPA time. For children with ID, object control skills were significantly related to their MVPA time, but in adolescents with ID, none of the skills were significantly related to MVPA time. Implementing FMS intervention programs focusing on object control skills in Chinese children with ID is recommended. The findings of this study can assist in evidence-based school practice, policy, and intervention design to increase MVPA engagement among children and adolescents with ID.

## Figures and Tables

**Figure 1 ijerph-19-13057-f001:**
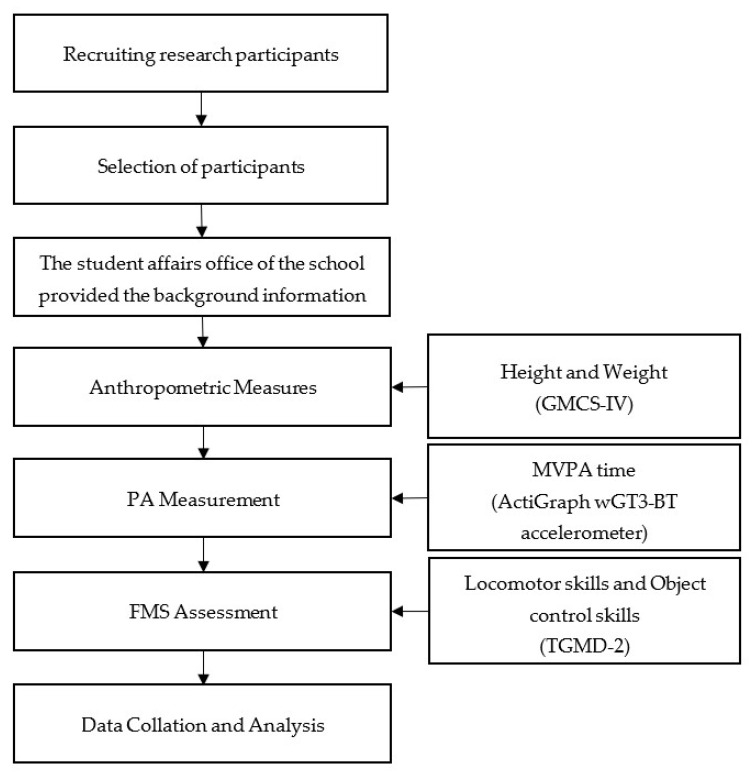
Study flow chart. Note: FMS, fundamental movement skills; MVPA, moderate-to-vigorous intensity physical activity; PA, physical activity.

**Table 1 ijerph-19-13057-t001:** Descriptive characteristics of the study sample.

Variables	Overall (*n* = 93) *M* (*SD*)	Boys (*n* = 63) *M* (*SD*)	Girls (*n* = 30) *M* (*SD*)
Age (years)	13.27 (3.35)	13.33 (3.48)	13.13 (3.09)
Children (7–12 y)	49	34	15
Adolescents (13–17 y)	44	29	15
Height (m)	1.50 (0.16)	1.51 (0.17)	1.46 (0.14)
Weight (kg)	45.26 (14.88)	47.02 (15.94)	41.47 (11.77)

Note. M, mean; SD, standard deviation.

**Table 2 ijerph-19-13057-t002:** Proportion of participants displaying mastery/near-mastery/poor proficiency of locomotor and object control subskills.

	Mastery % (*n*)	Near-Mastery % (*n*)	Poor % (*n*)
Locomotor skills			
Run	6.45 (6)	18.28 (17)	75.27 (70)
Gallop	21.51 (20)	19.35 (18)	59.14 (55)
Hop	1.08 (1)	18.28 (17)	80.65 (75)
Leap	10.75 (10)	12.90 (12)	76.34 (71)
Horizontal jump	6.45 (6)	12.90 (12)	80.65 (75)
Slide	5.38 (5)	48.39 (45)	46.24 (43)
Object control skills			
Strike a stationary ball	2.15 (2)	20.43 (19)	77.42 (72)
Stationary dribble	8.60 (8)	24.73 (23)	66.67 (62)
Catch	12.90 (12)	20.43 (19)	66.67 (62)
Kick	2.15 (2)	11.83 (11)	86.02 (80)
Overhand throw	0 (0)	4.30 (4)	95.70 (89)
Underhand roll	1.08 (1)	7.53 (7)	91.40 (85)

**Table 3 ijerph-19-13057-t003:** FMS scores and MPVA times for children and adolescents with ID.

Variables		Locomotor *M* (*SD*)	Object Control *M* (*SD*)	Total FMS Scores *M* (*SD*)	MVPA Min *M* (*SD*)
Gender					
	Boys	28.17 (11.49)	25.37 (7.80) **	53.54 (18.25)	41.20 (12.75)
	Girls	25.67 (12.36)	22.03 (10.93) *	47.70 (21.95)	36.64 (11.55)
Age	Children (7–12 y)	26.00 (11.97)	23.22 (8.74) *	49.22 (19.48)	40.49 (13.10)
	Adolescents (13–17 y)	28.89 (11.49)	25.48 (9.28) **	54.36 (19.60)	38.88 (11.89)
Overall	Observed	27.37 (11.77)	24.29 (9.01) ***	51.66 (19.59)	39.73 (12.50)
	Maximum	48.00 (0.00)	48.00 (0.00)	96.00 (0.00)	—
		*t* = −16.91, *p* = 0.000	*t* = −25.39, *p* = 0.000	*t* = −21.83, *p* = 0.000	—

Note. * significant differences between locomotor and object control at *p* < 0.05, ** significant differences between locomotor and object control at *p* < 0.01, *** significant differences between locomotor and object control at *p* < 0.001; M, mean; SD, standard deviation; FMS, fundamental movement skills; MVPA, moderate-to-vigorous intensity physical activity.

**Table 4 ijerph-19-13057-t004:** MANOVA of FMS and MVPA by gender and age.

	Gender					Age					Gender × Age			
	*SS*	*df*	*F*	*p*		*SS*	*df*	*F*	*p*		*SS*	*df*	*F*	*p*
Locomotor	138.878	1	0.996	0.321		195.535	1	1.403	0.239		2.652	1	0.019	0.891
Object control	247.384	1	3.159	0.079		43.184	1	0.551	0.460		136.314	1	1.741	0.190
Total FMS scores	756.971	1	1.990	0.162		422.501	1	1.111	0.295		100.937	1	0.265	0.608
MVPA time	414.608	1	2.660	0.106		71.527	1	0.459	0.500		29.499	1	0.189	0.665

Note: FMS, fundamental movement skills; MVPA, moderate-to-vigorous intensity physical activity.

**Table 5 ijerph-19-13057-t005:** Correlations (r) between MVPA and locomotor skills, object control skills, and FMS in participants with ID.

Variable	1	2	3
**1** MVPA min			
**2** Locomotor skills	0.503 ***		
**3** Object control skills	0.552 ***	0.775 ***	
**4** Total FMS scores	0.556 ***	0.957 ***	0.925 ***
**1** Boys’ MVPA min			
**2** Boys’ locomotor skills	0.446 ***		
**3** Boys’ object control skills	0.518 ***	0.783 ***	
**4** Boys’ total FMS scores	0.502 ***	0.964 ***	0.920 ***
**1** Girls’ MVPA min			
**2** Girls’ locomotor skills	0.610 ***		
**3** Girls’ object control skills	0.604 ***	0.776 ***	
**4** Girls’ total FMS scores	0.644 ***	0.949 ***	0.935 ***
**1** Children’s MVPA min			
**2** Children’s locomotor skills	0.474 **		
**3** Children’s object control skills	0.563 ***	0.765 ***	
**4** Children’s total FMS scores	0.544 ***	0.957 ***	0.919 ***
**1** Adolescents’ MVPA min			
**2** Adolescents’ locomotor skills	0.571 ***		
**3** Adolescents’ object control skills	0.572 ***	0.781 ***	
**4** Adolescents’ total FMS scores	0.605 ***	0.955 ***	0.930 ***

Note: ** *p* < 0.01, *** *p* < 0.001; FMS, fundamental movement skills; MVPA, moderate-to-vigorous intensity physical activity.

**Table 6 ijerph-19-13057-t006:** Regression analysis results of participants’ FMS and average daily MVPA.

Model	Predictors	*B*	Standard Error of *B*	95% CI of *B*	*β*	*t*	*R^2^*	Δ*R^2^*
1	(Constant)	41.909	5.547	[30.888, 52.929]		7.555 ***	0.041	0.019
	Gender	4.628	2.747	[−0.828, 10.085]	0.174	1.685		
	Age	−0.401	0.386	[−1.167, 0.366]	−0.107	−1.039 *		
2	(Constant)	28.96	4.96	[19.109, 38.82]		5.841 ***	0.371	0.342
	Gender	2.288	2.287	[−2.257, 6.833]	0.086	1.000		
	Age	−0.812	0.323	[−1.454, −0.171]	−0.217	−2.516 *		
	Locomotor skills	0.257	0.144	[−0.029, 0.542]	0.242	1.787		
	Object control skills	0.534	0.188	[0.160, 0.908]	0.385	2.840 **		

Note: * *p* < 0.05, ** *p* < 0.01, *** *p* < 0.001.

**Table 7 ijerph-19-13057-t007:** Regression of MVPA for gender-specific/age-specific participants, including predictors, and adjusted.

Predictors	*B*	Standard Error of *B*	95% CI of *B*	*β*	*t*	*R^2^*	Δ*R^2^*
						0.328	0.294
Boys’ locomotor skills	0.113	0.190	[−0.268, 0.494]	0.101	0.591		
Boys’ object control skills	0.842	0.287	[0.268, 1.416]	0.515	2.936 **		
						0.486	0.427
Girls’ locomotor skills	0.472	0.221	[0.017, 0.926]	0.505	2.133 *		
Girls’ object control skills	0.198	0.248	[−0.312, 0.709]	0.188	0.799		
						0.328	0.284
Children’s locomotor skills	0.097	0.209	[−0.324, 0.519]	0.089	0.465		
Children’s object control skills	0.736	0.285	[0.162, 1.310]	0.491	2.582 *		
						0.378	0.331
Adolescents’ locomotor skills	0.377	0.214	[−0.055, 0.808]	0.364	1.763		
Adolescents’ object control skills	0.322	0.278	[−0.241, 0.885]	0.250	1.156		

Note: * *p* < 0.05, ** *p* < 0.01.

## Data Availability

The data collected and analyzed during the current study are available from the corresponding author upon request.

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
