# Peer review of "Associations between Fundamental Movement Skills and Moderate-to-Vigorous Intensity Physical Activity among Chinese Children and Adolescents with Intellectual Disability"

_ijerph, 2022, doi:10.3390/ijerph192013057_

Round 1

Reviewer 1 Report

Well done to the authors, I enjoyed reading this paper and the findings which came from the study. This is an interesting piece of research on a population that is understudied specifically on this topic of FMS and its relationship with PA. This paper is therefore of great importance.

The paper is well written and the results are clearly explained. I have made some suggestions to improve the paper further in the attached document. I think the decisions made around the accelerometer protocols require greater explanation in parts and you also need to acknowledge their limitations in greater detail in the concluding paragraph.

Reviewer 2 Report

Dear Author,

I suggest rethinking the construction of the study objective sentence. Assessing the relationship assumes that there is some kind of relationship between the variables of interest. Thus, I believe the phrase "assessing whether there are associations between FMS and moderate and vigorous PA" is more appropriate.

Instead of purposive sampling, I suggest convenience sampling, or non-probability sampling.

Since the study is very specific to a region in China, I recommend a brief explanation of how the school classifies children with intellectual disabilities, in order to present how well this classification is aligned with international standards.

Another point is to better present the different severities of intellectual disability. Was the adjustment made for more severe cases? If yes, how was this adjustment done? If not, I suggest indicating the lack of adjustment as one of the limitations of the research.

One point very much missed by the authors is the determinants of physical activity in children with ID. Do the distinct levels of ID worsening have different determinants?

Also, one point that I think is important: it is very difficult to do a study like yours. The population is very specific, which somehow justifies convenience sampling. In this way, how to think about a perspective of larger studies (or population studies) among people with ID? What are the obstacles and possibilities?

Reviewer 3 Report

First of all, I would like to thank you for giving me the opportunity to write an article with a very interesting theme. however, I must indicate some comments that I would like the authors to attend to.

General comments

Thank you so much for the opportunity to review this manuscript. This investigation fills in some gaps on understanding the effect of motor intervention on motor performance in children with intelectual disabilities.  El documento , en general, está bien escrito y es fácil de seguir. However, the rationale for the study, especially the gender effect, and age group was not clearly laid out. Several importance details on the methods were not described. Overall, the current manuscript needs more improvement and the current version is not publishable.

Tittle:

The title must indicate the region where the study was carried out.

Abstract

It is not necessary to indicate the subsections in the abstract. For example, remove “1 Bacground”; "two. Methods”…

Significant results should be noted as well as the p value in the abstract (lines 16-19).

Key Words

keywords should be avoided to coincide with parts of the title (for example fundamental movement skills” and others

Introduction

Lines 39-40: this statement needs a citation to support “An essential contributor to the amount of PA children and adolescents take part in is fundamental movement skill (FMS) competence

More current references than complete the introduction should be included.

Could you explain more about those studies that reported gender differences?

Material and methods

Authors are suggested to include a flow chart to improve monitoring of the research process.

The classification of the students by age (7-12) years and 13-17 is not very good since in the youngest

Line 159 That the application was carried out by the teachers of the centers could be a bias in the application of the accelerometers. Did all teachers apply it in the same way?

FMS Measures: How was good, fair, or poor performance on the measured skills quantified or assessed?

Statistic analysis:

The statistical package that was used for the analyzes must be indicated, as well as the confidence index and the margin of error. The normality test of the data is missing. And add the "Cohen's d" statistic for statistical power.

In any case, the statistical tests are not appropriate (t-test). The most appropriate test would be a MANOVA where a first factor was gender (boys vs. girls) and a second factor was age group (7-12 years and 13-17 years)

Results:

Line 197 (the results must be written and presented correctly: line 197: The authors write “(t = 0.268, P = 789)” the degrees of freedom are missing, the symbol “p” must be in lowercase and in italics, Its value it is incomplete, it lacks one = and one point (=.789) and these results must also be accompanied by the statistical power using Cohen's d.

Even so, it is considered that the results of the FMS must be completely redone because the statistical test that would be necessary to perform is not adequate. Otherwise, a very wide age range is being evaluated if it is only compared by gender (from 7 to 17 years). Biases occur.

Regarding the values of the MVPA, the same should be done as with the FMS.

Discussion

The discussion should be based on the results of the investigation. As the statistical tests and the results are not adequate. The discussion must be redone completely.

Conclusions

The conclusions do not fit the results that should appear in the investigation

It is recommended that the conclusions be made again based on the new results after the correct analysis.

The limitations section must go before the conclusions (lines 381-396)

References:

Only 34% of the references used in the study are from the last 5 years. Authors are encouraged to include more current references.

Some of the references used are incomplete (for example reference 64) This reference lacks the year of publication

I hope that my suggestions and comments will help improve this manuscript.

Best regards

Round 2

Reviewer 3 Report

.